# Motivators and barriers to the acceptability of a human milk bank among Malaysians

**Kalaashini Ramachandran**[1], **Maznah Dahlui**[1,2,3], **Nik Daliana Nik Farid**[1,4]*

**1** Faculty of Medicine, Department of Social and Preventive Medicine, University Malaya, Federal Territory of Kuala Lumpur, Kuala Lumpur, Malaysia, **2** Department of Research Development and Innovation, University Malaya Medical Center, Federal Territory of Kuala Lumpur, Kuala Lumpur, Malaysia, **3** Facultas of Public Health, Department of Administration and Health Policy, Airlangga University, Surabaya, Indonesia, **4** Faculty of Medicine, Centre for Population Health, University Malaya, Federal Territory of Kuala Lumpur, Kuala Lumpur, Malaysia

* daliana@ummc.edu.my

## Abstract

The World Health Organisation (WHO) recommends that all babies be exclusively breastfed, stating that donor milk is the next best alternative in the absence of the mother's own milk. Milk sharing takes many forms, namely wet nursing, co-feeding, cross-feeding, and a human milk bank (HMB). However, the establishment of a human milk bank is still not widely accepted and is a debatable topic because of religious concerns in Malaysia. The aim of this study is to determine the facilitators and barriers among Malaysians towards the acceptance of an HMB. A cross-sectional study with 367 participants was conducted; the participants answered an online-validated, self-administered questionnaire. Data on socio-demographic, knowledge on breastfeeding benefits, knowledge and attitude on HMB-specific issues were analysed in terms of frequency before proceeded with multiple logistic regression. The majority of the respondents were Muslim (73.3%), had completed their tertiary education (82.8%), and were employed (70.8%). Only 55.9% of respondents had heard of HMB, stating the internet as their main source of information, but many respondents were agreeable to its establishment (67.8%). Most respondents had a good score on knowledge of breastfeeding benefits and on HMB-specific issues (70% and 54.2%, respectively), while 63.8% had a positive attitude towards HMB. In the multivariate analysis, mothers with a good score on general knowledge of breastfeeding (AOR: 1.715; 95% CI 1.047–2.808) were more likely to accept the establishment of HMB, while being a Muslim was negatively associated with its establishment (AOR = 0.113, 95% CI 0.050–0.253). The study found a high prevalence of mothers who were willing to accept the establishment of HMB. By educating mothers on the benefits of breastfeeding, as well as addressing their religious concerns, the establishment of a religiously abiding HMB in Malaysia may be accepted without compromising their beliefs or the health benefit of donor milk.

**Data Availability Statement:** Authors are not sharing the raw data, only those reported in the manuscript due to ethical restrictions imposed by the ethics committee. To request for data, an

official email must be sent to nmrr@nmrr.gov.my (NMRR ID: NMRR-20-2481-56918).

**Funding:** The authors received no specific funding for this work.

**Competing interests:** No authors have competing interests.

## Introduction

In many cultures, breastfeeding and breast milk sharing have been fundamental human practices for many centuries. The benefits of breast milk have been well evidenced, and children who have been breastfed are healthier compared to other feeding methods. Breast milk is viewed as the best source of nutrition and energy for infants [1] and is widely recognised as the required biological fluid for optimal growth and development. Breastfed babies have better protection from infections such as respiratory and ear infections, show a lower prevalence of chronic diseases such as obesity and diabetes, and have a lower mortality rate [2–5]. The World Health Organisation (WHO) and American Academy of Paediatrics have stated that the ideal pattern of infant feeding is exclusive breastfeeding during the first six months of a child's life and the introduction of complementary health food after six months of age while continuing to breastfeed until two years of age or beyond [3,6]. Among premature infants, breast milk has been reported to reduce their health risk, including the risk of necrotising enterocolitis and late- onset sepsis [7,8]. The most recent report by the WHO has also noted that over 820,000 children could have been saved yearly if they were breastfed [9]; hence, the WHO has strongly recommended that low birth weight (LBW) infants should be given expressed breast milk from a donor mother when the mother's milk is not available [10].

Despite the many efforts by the WHO, national governments and non-governmental organisations (NGOs) to support and actively promote breastfeeding, some mothers suffer from lactation failure or perceived insufficient milk supply [11,12]. Lactation failure or the inability to produce breast milk is not a new problem, being documented in the earliest medical encyclopaedia, *the Papyrus Ebers*, which originated in Egypt; in the document, wet nursing was stated the main alternative feeding option [13]. In circumstances when the mother's own milk is not available for the infant, the WHO has recommended donor breast milk as the next best option when compared with formula feeding, especially among vulnerable babies [10]. In line with the study done by Quigley et al., formula-fed infants are at twice the risk of developing necrotising enterocolitis compared with infants fed by expressed donor's breast milk [14]. Many international studies also found that the use of donor milk is more cost effective, with substantial cost savings for the individual, communities, and healthcare systems by reducing the number of cases of necrotising enterocolitis, late onset sepsis, childhood food intolerance, and hospital length of stay [14,15].

Milk sharing has been described as a commerce-free practice where the donor mother directly feeds the recipient's child, or the donor gives the expressed breast milk to the recipient's child for feeding [16]. Milk sharing takes many forms, namely wet nursing, co- feeding, cross-feeding, and an HMB [17]. Historically, the concept of a wet nurse was a widely accepted practice among society and the preferred alternative feeding option before the innovation of the feeding bottle and manufacturing of formula milk [18]. Although "wet nursing" is slowly becoming less used [19], the practice of breast milk sharing as an alternative method of infant feeding is still strong either via an HMB or via an informal platform, namely social media, through internet-based or peer milk sharing.

As in many parts of the world, informal breast milk sharing is rapidly rising in popularity among mothers in Malaysia who are motivated and determined to feed their babies with breast milk but are unable to do so. A global network for milk sharing on Facebook called 'Human Milk 4 Human Babies' is actively supported by web users throughout Malaysia [20]. However, the establishment of breast milk sharing, and banking is a rather controversial issue and is still being debated in Malaysia because of the concept of 'milk kinship' in Islam [21]. According to the Department of Statistics, Islam is the most practiced religion in Malaysia [22], and for Muslim communities, the sharing of breast milk must be well handled to ensure the

inheritance of lineage for those participating in this practice. The practice of informal milk sharing, or wet nursing has certain rules in Islam known as *Hukum Tahrim*, which means a recipient child whose hunger is satisfied when breastfed from the same donor five times or more is considered as the donor's milk daughter or son, and this child's relationships with the donor's biological children becomes milk siblings which means that marriage is forbidden between them [23]. The issue has made formal breast milk sharing and the establishment of an HMB an ongoing debatable topic [21] leading to an informal milk-sharing practice utilising the online platform. Malaysia recently launched its first Syariah (Islamic Law) -compliant human milk bank [24] which basically denotes that the activity or services adhered to the principles of Islamic Law. However, because the milk bank in Malaysia is still in its infancy, the full extent of its function and operation is still unknown. To date, only one study has been conducted in Malaysia to evaluate the feasibility of breast milk donation as an alternative to HMB [25]; the study reported that 88% and 77% of the donors and recipients, respectively, were Muslims, thus concluding that breast milk donation is an acceptable alternative compared with HMB.

The practice of milk sharing through online platforms and milk sharing networks is not free of risk. There is the risk of bacterial transmission and viruses, drug and substance exposure as well as microbial contamination during the handling and storage processes [16,26]. This practice has received much attention in the scientific literature and among the public health community, which has highlighted the risk associated with milk sharing [27]. The American Academy of Paediatrics, the US Food and Drug Administrations, La Leche League International, Human Milk Banking Association of North America (HMBANA) and European Milk Bank Association (EMBA) have all expressed concern about the safety of informal breast milk sharing [28], stating it is safer to obtain milk from an HMB.

In light of the significant concern related to the safety of breast milk sharing and to fully comprehend its health benefit and that of an HMB, it is vital to first understand mothers' knowledge and attitude towards HMBs. It is also important to estimate the prevalence of mothers who are aware and agree to the establishment of an HMB. A study was conducted to determine the motivators and barriers among mothers towards the acceptance of an HMB in Malaysia. The findings can guide future public health programmes when promoting safe breast milk sharing and HMBs, as well as in utilising donor milk from a milk bank.

## Materials and methods

### Study design and data collection

A cross-sectional study via online surveys were conducted from June 2021 until November 2021. A self-administered validated questionnaire formatted as google forms were distributed via social media and WhatsApp targeting Malaysian mothers who were currently pregnant or those who were either breastfeeding or who had breastfed before were invited to participate.

Sample sizes were estimated based on a study by Keim et al (2014) on breast milk sharing attitude, whereby the lowest odds ratio was 3.43 for mothers who pumped milk to feed their child and the highest odds ratio was 3.7, which was the gestation age of the infant [26]. The power of the study was set at 80% with a 95% level of significance. Open Epi software was used to calculate the sample size. The resulting sample size was a minimum of 306, and an extra 20% was added to prevent incomplete data, resulting in a total sample size of 367 subjects.

The inclusion criteria were Malaysian mothers who were currently pregnant or mothers who were either breastfeeding or who had breastfed before, well versed in English and Malay, and those who provided consent. Non-Malaysian mothers, mothers who could not communicate in English or Malay, and mothers who did not provide consent were excluded. A

convenient sampling technique was used. The participation information sheet and consent form were also available online, and each participant was required to fill out both forms prior to answering the questionnaire. Each participant was required to complete all the questions in the self-administered questionnaire.

The instrument was partly developed and partly adopted from two open access journals because no suitable tool was available for local use. Emails were sent to the authors to inform them that part of their questionnaire would be adopted for the current tool. In the preliminary phase, 37 items were developed both through extensive literature searches and consultations with relevant experts [29–31]. The content of the online self-administered questionnaire was divided into to three domains, which included four parts. The domains of interest were socio-demographic factors, knowledge and attitude. In the first domain, the subjects were asked about their sociodemographic backgrounds. Questions were asked regarding the participant's gender, age, race, religion, marital status, education level, employment status and monthly income. Monthly household income was further divided during the analysis into income less than RM 5000, which included the B40 (below 40) category and equal or more than RM 5000, which included the M40 (middle 40) and T20 (top 20) categories, here according to the Income Classifications in Malaysia [22]. The second and third domains (the knowledge and attitude domain) were covered in the following parts of the questionnaire. The second part of the questionnaire was adopted from a study done by Radzniwan et al. (2009), in which the participants were asked about their general knowledge and practices of breastfeeding, the number of children that were breastfed and the duration of being breastfed for each child. In this part, there were 10 questions on general knowledge regarding breastfeeding. The participants were given the option of true, false and do not know. The questions asked were on when to initiate breastfeeding, the duration of exclusive breastfeeding and the different benefits of breastfeeding. The third part consisted of 10 questions on the participants' knowledge, attitude and behaviour regarding breast milk sharing which include the participants' opinions on breast milk sharing, willingness to become either a donor or recipient of shared breast milk from a known or unknown mother, issues associated with breast milk sharing and whether breast milk sharing is forbidden according to the religious concept. The participants' opinion on breast milk sharing was questioned using the definition from the study by Palmquist and Doehler (2016), in which the study defined breast milk sharing as the practice of a donor mother providing expressed breast milk to a recipient family for the intention of infant feeding or directly nursing a recipient infant [16]. Regarding issues on breast milk sharing, the participants were asked about issues concerning milk sharing, chances of getting infected during breast milk sharing (BMS) and infection control standards; the participants could provide more than one answer. The last part of the questionnaire was on the attitude and behaviour of mothers towards the usage of an HMB and consisted of eight questions. This part was adopted from a study by Ahmet Karadag et al. (2015) [31]. Those participants who were willing to accept the establishment of an HMB in Malaysia were categorised as HMB acceptance and those who did not as HMB reluctance.

Content validation of the questionnaire was assessed by five field experts. The questionnaire was then translated from English to Malay language via backward and forward translation by two schoolteachers who were well-accomplished in both languages. Next, a pilot test of the questionnaire was conducted among 30 intended respondents [32]. No issues were raised regarding the questionnaire or the understandability of the instrument. Additionally, all respondents took less than 30 minutes to answer the questionnaire. This was then proceeded with exploratory factor analysis (EFA) using factor analysis software and confirmatory factor analysis (CFA) using Smart PLS version 4 software to determine the reliability, dimensionality and validity of the instrument. The results revealed excellent content validity, and the

reliability of the knowledge domain and attitude domain were 0.900 and 0.869, respectively, suggesting good reliability of the questionnaire and sufficient discriminant validity. The questionnaire was then developed into a Google form and distributed online. The respondents were selected conveniently to be included in the analysis after fulfilling the inclusion and exclusion criteria.

## Data analysis

The independent variables included general knowledge of the benefits of breastfeeding, knowledge of breast milk sharing and HMBs, and attitudes towards breast milk sharing and HMBs. The dependent variable was the perceived acceptance of the establishment of an HMB in Malaysia. The background variables were age, religion, marital status, education level, employment status and household income. For HMB attitude and knowledge-related questions, correct and positive answers were given one mark for scoring without any negative marking involved. The total score for knowledge ranged from 0 to 18 and scores of more than 13 indicated good knowledge, while the total score for attitude ranged from 0 to 19 and scores of more than 10 indicated positive attitude towards HMB.

The completed Google questionnaires were exported from Microsoft Excel to SPSS for further analysis. Information that could identify individual participants were not accessible by the authors during and after data collection. The raw data were cleaned and coded accordingly for tabulation. Continuous data that were normally distributed were summarised in the mean and standard deviation, while categorical data were reported in frequencies and percentages. The independent variables were then determined for association with the acceptability to the establishment of an HMB using inferential analysis either by adopting Pearson's chi-square test or Fischer's exact test with statistical significance set at $p < 0.25$. Variables with $p < 0.25$ were then included in the multivariate binary logistic regression model to identify statistically significant predictor variables. The adjusted odds ratio (AOR) was computed, and $p < 0.05$ was statistically significant. The results were then presented as either tables or graphs.

## Ethics approval

Ethical approval was obtained from the National Medical Research Register (NMRR- 20–2481–56918 (IIR)) and the University of Malaya Research Ethics Committee (UMREC) (UM. TNC2/UMREC– 946). Eligible participants who were voluntarily willing to participate and fulfilled the inclusion criteria were included after providing written consent at the beginning of the questionnaire, which was distributed online. The study was conducted in compliance with the ethical principles outlined in the Declaration of Helsinki and the Malaysian Good Clinical Practice Guideline.

## Results

A total of 367 respondents were included in the analysis, of whom all consented and completed the questionnaire, with no missing data. The ages of the respondents ranged from 20 to 49 years old, with the majority being less than 35 years of age, as shown in Table 1. The average age of the respondents was 31.76 ± 4.44 years. The majority of the respondents were Muslim, had successfully completed their tertiary education, and were employed (73.3%, 82.8% and 70.8%, respectively). Furthermore, almost half of the respondents (51%) had a household income of ≥ RM 5000. The majority had satisfactory knowledge of the benefits of breastfeeding, with 257 out of 367 (70%) showing good scores (score 8 or more out of 10). This is shown in Table 2.

**Table 1. Sociodemographic characteristics of the respondents (n = 367).**

| Variables | Mean (SD) | n | % |
|---|---|---|---|
| Age | 31.76 (4.441) | 367 | 100.0 |
| Ethnicity Malay Chinese Indian | | 269 | 73.3 |
| | | 55 | 15.0 |
| | | 36 | 9.8 |
| Other | | 7 | 1.9 |
| Religion Islam Buddhist Hindu Christian Other | | 269 | 73.3 |
| | | 55 | 15.0 |
| | | 36 | 9.8 |
| | | 7 | 1.9 |
| | | 0 | 0.0 |
| Marital status Single Married | | 0 | 0.0 |
| Divorced/widow | | 366 | 99.7 |
| | | 1 | 0.3 |
| Education level Tertiary education Secondary education | | 304 | 82.8 |
| Primary or no education | | 63 | 17.2 |
| | | 0 | 0.0 |
| Employment status Employed Unemployed | | 260 | 70.8 |
| | | 107 | 29.2 |
| Household income (RM) | | | |
| < RM 5000 | | 180 | 49.0 |
| ≥ RM 5000 | | 187 | 51.0 |

As seen in Table 3, approximately 60.5% of the mothers provided the correct definition of breast milk sharing based on their understanding and knowledge, while 10.1% gave the wrong definitions. However, about 29.4% (n = 108) answered 'not sure'. At a personal level, the majority of the participants were willing and optimistic of breast milk sharing practices from a known donor or recipient, as seen in the Figs 1 and 2 below. When the identity of the donor or recipient was anonymous, more mothers (52.0%) were willing and not selective to donate milk when the recipient baby was unknown, and only 25.3% of the mothers were willing to feed their baby from an unknown donor. Nearly all the respondents (99.2%) responded positively

**Table 2. General knowledge characteristics of the respondents towards breastfeeding.**

| No | Questionnaire item | True n (%) | False n (%) | No idea n (%) |
|---|---|---|---|---|
| 1 | A woman who is fully breastfeeding is less likely to become pregnant three months after delivery compared with women who formula feed | 279(76.0) | 60 (16.3) | 28 (7.6) |
| 2 | A breastfed baby is protected against gastrointestinal infection | 341(92.9) | 8(2.2) | 18(4.9) |
| 3 | A breastfed baby is not protected against allergies | 200(54.5) | 134(36.5) | 33(9.0) |
| 4 | Breast feeding should be initiated immediately after birth | 364(99.2) | 1(0.3) | 2(0.5) |
| 5 | Frequent breast feeding in the early newborn period can help reduce jaundice | 350(95.4) | 9(2.5) | 8(2.2) |
| 6 | Mothers intending to breastfeed shouldexpect sore nipples as a normal part of breastfeeding | 301(82.0) | 49(13.4) | 17 (4.6) |
| 7 | Excusive breastfeeding should be practiced for the first six months | 351(95.6) | 13(3.5) | 3(0.8) |
| 8 | Breast milk and formula milk provide the same health benefits for infants | 35(9.5) | 320(87.2) | 12(3.3) |
| 9 | Colostrum is good for baby | 328(89.4) | 10(2.7) | 29(7.9) |
| 10 | Breastfeeding should be stopped if either/both the baby/mother is sick | 123(33.5) | 198(54.0) | 46(12.5) |

**Table 3. Knowledge and attitude of the participants about breast milk sharing.**

| No | Questionnaire item | True/Permissible/ Yes n (%) | False/ Forbidden /No n (%) | No idea/ Not sure n (%) |
|---|---|---|---|---|
| 1 | Correct definition of BMS | 222 (60.5) | 37(10.1) | 108 (29.4) |
| 2 | Would you feed your baby with the breast milk of another woman you know when you do not have sufficient breast milk? | 279 (76.0) | 88(24.0) | |
| 3 | Would you feed your baby with the breast milk of another woman whom you do not know when you do not have sufficient breast milk? | 93 (25.3) | 274(74.7) | |
| 4 | Would you give the breastmilk that your baby does not need to some other baby you know? | 331 (90.2) | 36 (9.8) | |
| 5 | Would you give the breastmilk that your baby does not need to some other baby you do not know? | 191(52.0) | 176(48.0) | |
| 6 | In a Muslim country, what is your opinion when a baby whose mother has no breastmilk is breastfed by another woman? | 364(99.2) | 3(0.8) | |
| 7 | Does feeding a baby from another woman's milk with a bottle establish a foster motherhood between the baby and women? | 361(98.4) | 6(1.6) | |
| 8 | If two different babies are breastfed by the same mother, do they become foster siblings? | 354(96.5) | 13(3.5) | |
| 9 | What is your opinion about the marriage of foster brothers and sisters? | 20(5.4) | 347(94.6) | |
| 10 | Issues that could occur during breast milk sharing: Milk kinship Chances of infection Infection control standards | 286(77.9) 94(25.6) 109(29.7) | 81(22.1) 273(74.4) 258(70.3) | |

regarding the acceptance of breast milk sharing when the need arose in a Muslim country. A large percentage of them acknowledged the establishment of foster motherhood between the donor mother and recipient babies (98.4%), establishment of foster siblings between the donor mothers' children and recipients' babies (96.5%) and forbidden marriages between foster

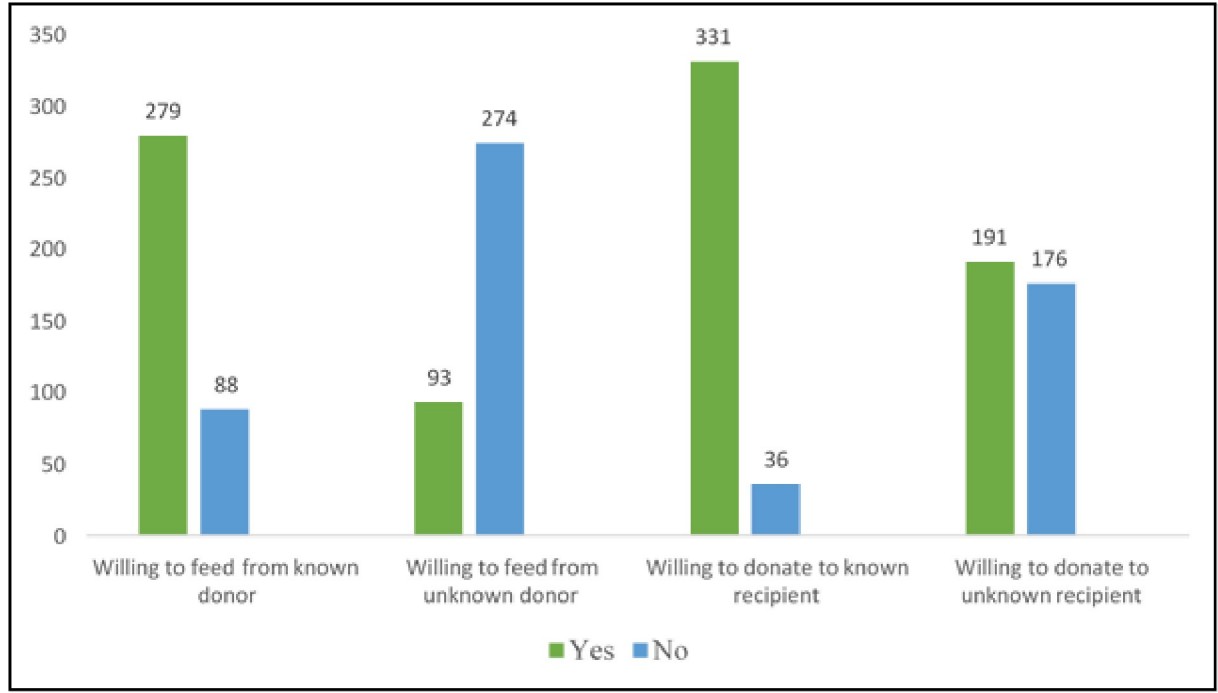

**Fig 1. Respondents' willingness to become a donor or recipient in BMS practices.**

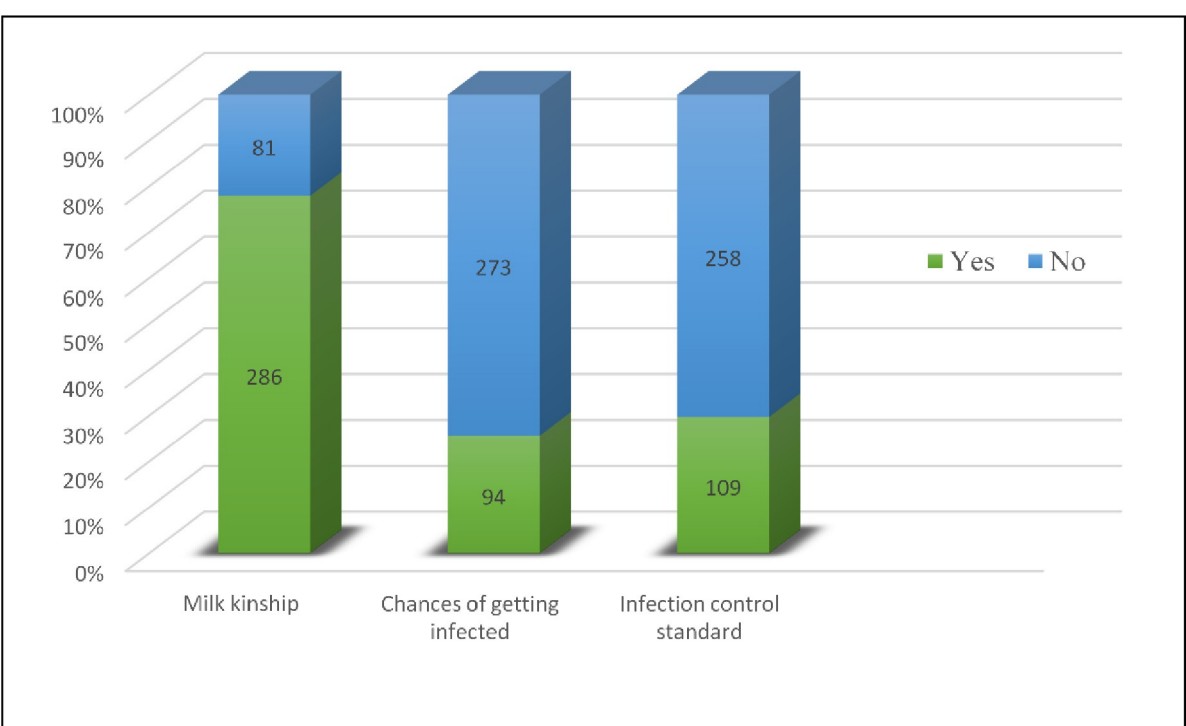

**Fig 2. Participants' concerns regarding what could occur during BMS.**

siblings (94.6%). When asked about the issues that could occur during breast milk sharing, which was on a question where the respondents could select more than one response, the majority answered milk kinship (n = 286). The chances of getting infected or infection control standards were not a major concern.

Table 4 summarises the knowledge and attitude of the respondents towards an HMB. More than half of the respondents (55.9%) had heard of an HMB and of those, social media (29.4%) and internet (29.4%) were listed as their main sources of information, followed by friends and relatives (13.4%), health practitioners (12.0%) and newspapers (4.6%). From the responses to the questionnaire, most mothers (88.8%) would prefer if the donor identity was revealed, while the education background of the donor was only of concern to less than half of them (45.2%). Almost all the respondents agreed that the probability of cross-contamination (94.8%) and donor health status (98.9%) were factors to be focused on during the establishment of an HMB. A higher proportion of mothers were not in favour when asked on the need to compensate donor mothers and impose service charges by the milk bank (61% and 57.5%, respectively). Despite the possible concerns, the authors noticed that almost two-thirds of the respondents (67.8%) agreed to the establishment of an HMB in Malaysia.

The present study found that approximately a little more than half of the respondents (54.2%) had good knowledge of breast milk sharing and human milk bank–specific issues. This can be seen in Table 5, whereas a larger proportion (63.8%) answered positively for all the attitude-related questions, as shown in Table 6 below.

As shown in Table 7, the factors associated between the sociodemographic characteristics, scoring of the respondents' general knowledge on breastfeeding, scoring of the respondents' knowledge and attitude towards HMB and its perceived acceptance have been tabulated. All the independent variables with a level of significance of less than or equal to 0.25 were then included in the multivariate analysis.

**Table 4. Knowledge and attitude of the respondents towards the human milk bank.**

| No | Questionnaire item | True/Yes n (%) | False/No n (%) | No idea n (%) |
|---|---|---|---|---|
| 1 | Have you ever heard about a human milk bank? | 205 (55.9) | 162(44.1) | |
| 2 | The sources of this information? | | | |
| | a) Internet | 108 (29.4) | 259 (70.6) | |
| | b) Social media | 108 (29.4) | 259 (70.6) | |
| | c) Newspaper/magazine | 17 (4.6) | 350 (95.4) | |
| | d) Friends/relative/neighbours | 49 (13.4) | 318 (86.6) | |
| | e) Health practitioner | 44 (12.0) | 323 (88.0) | |
| 3 | Should the identity of the donor be revealed? | 326 (88.8) | 41 (11.2) | |
| 4 | Is the educational background of the donor important? | 166 (45.2) | 200 (54.5) | 1 (0.3) |
| 5 | Factors that need to focus during the establishment of an HMB: | | | |
| | The probability of cross- contamination | 348(94.8) | 19 (5.2) | |
| | Donor health status | 363 (98.9) | 4 (1.1) | |
| | Person in charge of the operation | 275 (74.9) | 92 (25.1) | |
| 6 | Do you think that mothers who donate their breast milk to an HMB should be given money? | 143 (39.0) | 224 (61.0) | |
| 7 | Do you think that an HMB should charge money for the services (in terms of manpower and administration) they provide? | 156 (42.5) | 211(57.5) | |
| 8 | Do you agree on the establishment of an HMB in this country? | 249 (67.8) | 118 (32.2) | |

The respondents from an Islamic faith background (OR:0.109, 95% CI: 0.05–0.245) and from a household income of RM 5000 or less (OR: 0.418, 95% CI: 0.266–0.657) were less likely to accept the establishment of an HMB. Age, marital status, education level or employment status from the sociodemographic characteristics were found to have no significant effect on the decision to accept the establishment of an HMB. Next, in examining the scoring on the general knowledge of breastfeeding among the respondents, a good score was positively associated with the mother's agreement to establishing an HMB (OR: 1.847, 95% CI: 1.16–2.95). Finally, good or poor knowledge did not show any significant association with the respondents' perceived acceptance of the establishment of an HMB. A similar result was also seen among scoring the respondents' attitudes towards HMBs.

Independent variables with a significant *p*-value of 0.25 and less were included in the multivariable logistic regression analysis. Binary logistic regression analysis was performed to assess the relationships between the factors (independent variables) and perceived acceptance of the establishment of an HMB. All assumptions for the regression model were met.

All variables were first analysed using the enter method and were explored. The -2- likelihood ratio was also set as the critical value in the chi square table in the process of obtaining the significant factors. Next, the statistically significant final model containing the remaining factors are shown in the above tables. Hosmer and Lemmeshow test was performed for all the models, and all of them had a value of p > 0.05, indicating that the model fit was good. In addition, a backward likelihood ratio was also performed, and the results were compared,

**Table 5. Scoring of the participants' knowledge on breast milk sharing and human milk bank–specific issues (n = 367).**

| Scores | n (%) |
|---|---|
| Good (14–18) | 199 (54.2) |
| Poor (0–13) | 168 (45.8) |

**Table 6. Scoring of participants' attitudes towards breast milk sharing and human milk bank–specific issues (n = 367).**

| Scores | n (%) |
|---|---|
| Positive (11–19) | 234 (63.8) |
| Negative (0–10) | 133 (36.2) |

producing similar findings. Considering the parsimonious and good fit fulfilling all the assumptions of the regression model and that all the variables were clinically important, the author found this is the best model to be used.

The present study has identified that good knowledge on the benefits of breastfeeding is a significant predictor of the acceptance of the establishment of a, HMB (OR: 1.7515, 95% CI: 1.047–2.808). It was also observed that the Islamic religion was a significant barrier to the acceptance of an HMB in Malaysia (OR: 0.113, 95% CI 0.050–0.253).

## Discussion

Breastfeeding and breast milk sharing have been practiced in many cultures for many centuries. Reservations regarding the potential risk of feeding an infant with donor human milk provided by mothers who have not been systematically screened by a qualified health professional have sparked debate around milk sharing [33,34]. Thus, the establishment of an HMB and usage of pasteurised donor human milk is a safer option compared with the informal method of breast milk sharing. There are approximately 756 human milk banks operating in 66

**Table 7. Association between the factors and respondent's agreement to the establishment of an HMB in Malaysia.**

| Variables | Agreement to establish a HMB in Malaysia | | | | | | | | |
|---|---|---|---|---|---|---|---|---|---|
| | Total | Yes | No | p-value | Crude OR | 95% CI | p-value | AOR | 95% CI |
| Age<br>≤35<br>>36 | 288<br>79 | 193 (67.0)<br>56 (70.9) | 95 (33.0)<br>23 (29.1) | 0.514 | 0.834 | 0.484–1.438 | - | - | - |
| Religion    Islam<br>Non-Islam | 269<br>98 | 158 (58.7)<br>91(92.9) | 111(41.3)<br>7 (7.1) | **0.001** | 0.109 | 0.049–0.245 | **0.001** | 0.113 | 0.050–0.253 |
| Marital status Married<br>Single/divorced/widow | 366<br>1 | 248(67.8)<br>2(100.0) | 118(32.2)<br>0(0.0) | 0.678 | 0.678 | 0.631–0.727 | - | - | - |
| Education level Tertiary education<br>Primary/secondary/No education | 304<br>63 | 209 (68.8)<br>40 (63.5) | 95 (31.3)<br>23 (36.5) | 0.416 | 1.265 | 0.717–2.231 | - | - | - |
| Employment status Employed Unemployed | 260<br>107 | 176 (67.7)<br>73 (68.2) | 84 (32.3)<br>34 (31.8) | 0.921 | 0.976 | 0.602–1.582 | - | - | - |
| Household income (RM)<br>< RM 5000<br>≥ RM 5000 | 180<br>187 | 105 (58.3)<br>144 (77.0) | 75 (41.7)<br>43 (23.0) | **0.001** | 0.418 | 0.266–0.657 | 0.253 | 0.747 | 0.453–1.232 |
| Score on general knowledge on breastfeeding Good<br>Poor | 257<br>110 | 185 (72.0)<br>64 (74.6) | 72 (28.0)<br>46 (35.4) | **0.009** | 1.847 | 1.158–2.945 | **0.032** | 1.715 | 1.047–2.808 |
| Scoring of knowledge of BMS and HMB Good<br>Poor | 199<br>168 | 134 (67.3)<br>115 (68.5) | 65 (32.7)<br>53 (31.5) | 0.820 | 0.950 | 0.612–1.475 | - | - | - |
| Scoring of attitudes towards BMS and HMB Positive<br>Negative | 234<br>133 | 157 (67.1)<br>92 (69.2) | 77 (32.9)<br>41 (30.8) | 0.682 | 0.909 | 0.575–1.437 | - | - | - |

countries aim at improving neonatal health by making safe and pasteurized donor human milk available [35]. Turkish Ministry of Health proposed a pilot program for the establishment of a HMB in an Islamic country but unfortunately was later discontinued due to religious objections [31] until recently when Malaysia launched its first Syariah compliant HMB [24] almost 100 years after the establishment of the world's first HMB [36]. However, since the milk bank in Malaysia is still in its infancy stage, the full extent of its function and operation is still unknown to the researcher. Information on this HMB have not been widely and publicly disseminated and therefore not popularized within the community yet.

To the best of the authors' knowledge, this is one of the first studies conducted in Malaysia to determine the factors influencing the acceptability of the establishment of an HMB. Overall, in the present study, it was found that more than half of the participants (60.5%) were aware of breast milk sharing, which was reported as being higher compared with the studies done in either underdeveloped or developing countries [37–40]. We also found that more than half of the participants (55.9%) had heard of an HMB, and surprisingly, this result was marginally higher than studies done in other Asian countries reporting, at best, 20% of the participants in China and 49% of the participants in South Korea being aware of an HMB [41, 42].The prevalence in the present study was also significantly higher when compared with other studies conducted among majority Muslim participants like in Turkey, Bangladesh and Ethiopia, where each reported that almost all of the respondents surveyed never heard of an HMB or had a low level of awareness of an HMB [43–46].

However, the present study showed that more than half (67.8%) of the mothers were willing to accept the establishment of an HMB in Malaysia, though this result was still relatively low when compared with a recent international study conducted among mothers reporting that more than 77.6% of them accepted the establishment of an HMB in their country [47]. The differences in the prevalence of HMB acceptance may be attributed to the different study populations from different backgrounds and settings (i.e., Muslim vs. non-Muslim, community vs. hospital or clinics, developed vs. developing countries, urban vs. rural areas).

Informal milk sharing has been a common practice among close friends and family members for many generations. For Muslims, wet nursing is permissible when the mother is not able to produce breast milk; the Prophet Muhammad himself was breastfed by two different wet nurses [48]. Parents can mutually agree to employ wet nurses to feed their children when the need arises [49], hence demonstrating the preference of feeding babies with human milk instead of animal milk in Islam. Additionally, despite being an Islamic nation, breast milk sharing was reported to not be an issue within the Malaysian society and the rise in milk sharing practices is to a great extent attributed to the growing public's knowledge of the value and benefits of breast milk [23]. For these reasons, the prevalence of BMS awareness was higher compared with the findings from previous studies conducted in many developing countries [37–40]. Contrary to the awareness of breast milk sharing, the concept of an HMB may still be unfamiliar to or unpopular in Malaysia, especially without the public support or promotion by the religious committee such as Department of Islamic Development Malaysia (JAKIM).

The increasing demand for donated breast milk has created the necessity to establish an HMB, but religious reservation, traditional myths and maternal fear have certain contributions to the reluctance to accept an HMB in Malaysia, as demonstrated in studies from other countries [37,47]. Nonetheless, it should be acknowledged that mothers with knowledge of BMS and HMBs have a tendency to accept the establishment of one in Malaysia because of the benefits they have for vulnerable and preterm babies. Significant global variation in milk sharing and milk banking awareness are still considerably low in some nations, including Malaysia, where milk sharing, and milk banking knowledge and culture may be unfamiliar or not publicly promoted.

The majority of the participants in the current study obtained information regarding human milk banking from either the internet or social media, and only a small portion of them had heard about human milk donation from healthcare practitioners. This outcome is in line with other studies that reported that the source of information on BMS and HMBs was less likely to come from healthcare workers [26,41,50]. It can be reasoned that the public's knowledge of breast milk sharing has increased since 2010 with the emergence of online milk sharing networks that provide virtual spaces for mothers with surplus milk to share with mothers in need [51]. This finding also suggests that healthcare workers might lack knowledge of human milk sharing safety and resources and HMB usage, which needs to be rectified first. In countries where HMBs have been well established, a lack of discussion with families about milk donation and the function of donor human milk contributed to the limited knowledge and usage of human milk banks in the community [52]. Additionally, many international organisations like the American Academy of Paediatrics (APA) and HMBANA do not encourage milk sharing outside a certified human milk bank [53,54], which may contribute to the reluctance of HCWs to discuss milk sharing with parents. Many international studies have proven that the majority of parents who have health-related doubts tend to seek additional information from their doctors [55]. Thus, information from the health care professional is one of the key influencing factors for parents when making decisions for their child. Investing in educating healthcare workers on the benefits of donor human milk and HMBs can increase awareness among the community and lead to the acceptance of an HMB in Malaysia.

In the present study, 54.2% of the respondents achieved good scoring on the knowledge on HMB-related issues, while 63.8% answered positively on the attitude-related questions in the current study. No significant association was found between these two domains and the acceptability of an HMB; having good knowledge or a positive attitude towards an HMB did not play a vital role in increasing the acceptance of the establishment of an HMB in Malaysia. This finding differs from the study by Iloh et al., who found that mothers with adequate knowledge of the concept of breast milk sharing and HMBs were 10.8 times more likely (OR 10.76, 95% CI: 2.78, 23.67) to participate in breast milk sharing [56]. Another quantitative study conducted in China also reported a positive association between knowledge and acceptability towards HMBs among the research participants [41]. In the current study, no significant association between knowledge and attitude towards the acceptability of BMS among mothers was attributed to the absence of a fully functioning HMB in Malaysia. This explanation was affirmed by one study done in Uganda, which showed that no significant findings were seen between knowledge and awareness of HMBs because there were no milk banks in Uganda during the duration of that study [47]. In addition, the present study also found that almost 70% of the participants with good scores on breastfeeding knowledge were more likely to accept the establishment of an HMB in Malaysia (OR 1.72, 95% CI: 1.05, 2.81). This shows the high percentage of mothers who positively accept the establishment of an HMB in this cohort. Nonetheless, knowledge is an important factor during health promotions for HMBs, as proven by a study where the majority of the surveyed mothers reported that they had never heard of an HMB, and when they were provided with adequate and relevant information, most wanted the facility in Turkey and were willing to donate, if necessary [43]. One possible explanation for this is that delivering adequate and appropriate knowledge through the right mode of communication on the benefits and necessity of an HMB might influence acceptance among mothers (and family members). The Ministry of Health, through the Malaysian Breastfeeding policy, has made great strides to vigilantly promote and encourage breastfeeding in line with the WHO's vision [57], but the current study did not weigh breast milk sharing practices or collect any formal or adequate data regarding milk sharing practices in the community. Despite the absence of an HMB in Malaysia, the community has been practising breast milk sharing via

the informal platform on Facebook called 'Human Milk 4 Human Babies', which is actively supported by web users [20]. Informal breast milk sharing occurs in the community signifies that it is not a completely new concept, thus creating the platform to boost advocacy messages for HMBs. One way forward to boost HMB acceptance is to incorporate knowledge on safe breast milk sharing in existing breastfeeding programmes and educate mothers on the benefits of donor breast milk; here, an HMB is a paramount public health initiative that can help increase the awareness and acceptability of HMBs in the community.

The present study also reported that attitudinal variables, namely the probability of cross-contamination, donor's health status and the person in charge of the operation, were some of the main issues of concern that need to be given attention to during the establishment of an HMB. This finding was also documented in a few other studies where the positive acceptance towards milk banking was not without apprehension and where safety measures involved could be highly influential towards the mothers' decision to use the donor human milk from the HMB [50,58,59]. Prior to the establishment and launching of an HMB in Malaysia, addressing and ensuring the safety measures throughout the collection, pasteurisation, storage, handling and dispensing process of human milk is crucial to gaining the public's confidence.

By using internationally recognised hazard management approaches such as the Hazard Analysis and Critical Control Point (HACCP) released by PATH to ensure safety and quality is necessary if Malaysia plans to establish a milk bank that is accepted by the public. PATH's global HMB implementation framework, which is based on input and tools from internationally established HMBs, has defined the basic requirements and quality principles that should be followed by all milk banks [60]. Mothers should be made aware that a milk bank ensures that donated breast milk goes through a rigorous process of screening, pasteurisation and hygienic handling and storage, hence making donor milk from the milk bank safe for infants who do not have access to their own mothers' milk (unlike informal milk sharing and wet nursing). Transparency of this process can help in gaining the trust of mothers and their family members to ensure infants get the best nutrition for optimal growth.

It should be acknowledged that Islam is Malaysia's largest religion. Religion was found to be a significant barrier to the acceptance of an HMB in this cohort, as highlighted in other studies [43,61,62]. The majority of the participants in the present study reported that milk kinship was an important issue of concern during breast milk sharing. The anonymity and multiplicity of donors to the milk bank contributed to their reluctance to use the milk from an HMB [63]. Knowledge and understanding the rationale of milk kinship can guide healthcare providers and stakeholders to develop a plan of action that will increase the confidence and acceptance among Muslim mothers in the establishment of a culturally and religiously abiding HMB in Malaysia without compromising their belief or the health benefit of donor milk.

Developing a registry or a donating system database within the milk banking institution where the identities of both the donor and recipient are voluntarily documented and easily assessable by families who need donor milk can reduce the anonymity issues with conventional milk banks. Additionally, parents should also be given the option to milk kinship, and consent should be taken from both the donor and recipient families. Pathways to distributing donor milk to recipient babies from families who are not keen on milk kinship should be different from families agreeable to milk kinship. Families who refuse milk kinship should also not receive more than three feedings from a donor mother, and the system should have a warning sign to indicate this. Transparency can improve the confidence level towards milk sharing and milk banking, especially among Muslim parents. The awareness and receptivity among Muslim parents can also be improved by involving religious leaders, beginning with the process of formulating Syariah-compliant guidelines to educate the public on the permissibility of using donor's milk, if necessary, to save an infant's life.

Furthermore, the religious department could be in charge of monitoring the religious regulations in the milk banks and ensure that there are no mismanagements to protect the donors or recipient's lineage. Additionally, engaging the religious department to monitor the cleanliness as well as the method of both storing and handling the donated breast milk will contribute the public's confidence level since the concept of 'halal' or permissible in the Quran includes these domains [64]. Proximal determinants such as maternal age, education level or household income did not influence the acceptance of a human milk bank in the present study, which was also echoed by another study done by Meneses et al. (2017) to determine the prevalence and factors associated with the donation of human milk. [65]. The majority of the participants in the present study did not think it was necessary to compensate the donors, which is similar to the findings from other countries, where the primary reasons for mothers to donate their breast milk were both altruistic and to save a child in need rather than monetary gain [59,66]. Although the motivation of donor mothers remains subjective, reimbursing donor mothers can be a financial option, especially those from the lower income group. There is the possibility that milk for an infant of a lower income donor mother could potentially be diverted to meet incentivised requests for donor human milk. Other concerns include the desire to enhance profits by diluting or adding additives to milk, hence compromising the nutrient quality and safety of donor human milk. More than half of the participants also believed that the recipients should not be charged for the services provided by the milk bank for their administrative costs or human resources. Establishing a human milk bank under the support and management of the government without any financial compensation to the donor can avoid exploiting donor human milk and ensure the sustainability of an HMB.

## Strengths and limitations

To the authors' knowledge, this is the first study on the knowledge, attitude and acceptability of the establishment of an HMB in Malaysia. The relatively new interest in HMBs and religiously sensitive topics have made it a challenge for other researchers to gain information; therefore, the results from the current study have presented valuable information on the acceptability of HMBs in Malaysia. The findings can be a guide for future studies and for stakeholders and policy makers during health promotional activities as they create related policies.

However, the present study was based on each participant's self-perception at the given time. This perception may change over time, place and context and might produce a different result in a different setting. Selection bias was another limitation. The sampling technique used was convenient sampling because of the recent pandemic, which may have influenced the type of participants recruited. It should also be noted that many quantitative respondents were recruited from social media breastfeeding groups, affecting their level of knowledge and pre-existing confidence, making them more receptive to novel interventions like an HMB. Attempts were made to improve the convenience sampling which included obtaining the miniature version of the intended population to improve the representativeness of this study, questionnaires were distributed via diverse platforms and at different timing allowing equal opportunity for the public to answer the questionnaire, as well as using a larger sample size in this study. Furthermore, mothers from rural areas and those without access to technology may not have been included because it may be difficult to capture their participation. Finally, there might also be a component of social desirability because the participants may be afraid of the stigma arising from their answer on this religiously sensitive topic; therefore, the participants were informed and given reassurance of their confidentiality and the level of anonymity. The study's ethnic composition is comparable to Malaysia's general ethnic structure [22], however, it is important to carefully interpret the results of this study to the general population with caution.

## Conclusion

In conclusion, the present study has found a high prevalence rate of mothers (67.8%) who were willing to accept the establishment of an HMB in Malaysia. The main motivator towards the acceptability of an HMB was good knowledge of the benefit of breast milk and breastfeeding, while a significant barrier was related to religion reservation. Hence, prior to promoting an HMB in Malaysia, there should be a continuous investment in educating mothers on the benefits of breastfeeding and providing accurate evidence-based information to mothers on HMBs. In addition, efforts to address the religious concerns of Muslims should be made so that the establishment of a religiously abiding HMB in Malaysia may be accepted without compromising their belief or the health benefit of donor milk. The results from the present study can help guide and build healthy public policy on HMBs.

## Supporting information

**S1 Checklist. STROBE statement—checklist of items that should be included in reports of observational studies.**
(PDF)

## Acknowledgments

The authors would like to express their deepest gratitude to all the participants who participate in this study. We would also like to thank the experts for validating our questionnaire and Prof Dr Mahmood Danee for guiding us with the validation analysis.

## Author Contributions

**Conceptualization:** Kalaashini Ramachandran, Maznah Dahlui, Nik Daliana Nik Farid.

**Data curation:** Kalaashini Ramachandran.

**Formal analysis:** Kalaashini Ramachandran.

**Investigation:** Kalaashini Ramachandran.

**Methodology:** Kalaashini Ramachandran, Maznah Dahlui.

**Supervision:** Maznah Dahlui, Nik Daliana Nik Farid.

**Validation:** Maznah Dahlui.

**Writing – original draft:** Kalaashini Ramachandran.

**Writing – review & editing:** Kalaashini Ramachandran, Maznah Dahlui, Nik Daliana Nik Farid.

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
