## [Decision Letter · Decision Letter 0]

8 Feb 2024

Motivators and barriers to the acceptability of a human milk bank among Malaysians

PONE-D-23-14900

Dear Dr. Nik Farid,

We’re pleased to inform you that your manuscript has been judged scientifically suitable for publication and will be formally accepted for publication once it meets all outstanding technical requirements.

Kind regards,

Nneoma Confidence JeanStephanie Anyanwu, Ph.D.

Academic Editor

PLOS ONE

*Journal requirements:*

1. We note that your Data Availability Statement is currently as follows: “All relevant data are within the manuscript and its supporting information files.”

Reviewers' comments:

Reviewer's Responses to Questions

**Comments to the Author**

*Please note that Reviewer #1 was invited twice in error, and is the same researcher as Reviewer #3.*

1. Is the manuscript technically sound, and do the data support the conclusions?

Reviewer #1: Yes

Reviewer #2: Yes

Reviewer #3: Yes

2. Has the statistical analysis been performed appropriately and rigorously? 

Reviewer #1: Yes

Reviewer #2: Yes

Reviewer #3: Yes

3. Have the authors made all data underlying the findings in their manuscript fully available?

Reviewer #1: Yes

Reviewer #2: Yes

Reviewer #3: Yes

4. Is the manuscript presented in an intelligible fashion and written in standard English?

Reviewer #1: Yes

Reviewer #2: Yes

Reviewer #3: Yes

5. Review Comments to the Author

Reviewer #1: I have no comments and it is well writen. This subject is very important for breastfeeding problems.

xxxxxxx xxxxxxxxx xxxxx xxxxxxxxxxxxxxxxxxxxxxxxxxxxxxxxxxxxxxxxxxxxxxxxxxxxxxxxxxxxxxxx

Reviewer #2: Dear Editor of the Journal PLOS ONE,

Greeting!!!

Thank you for sending me the manuscript Ref: Submission ID PONE-D-23-14900, entitled "Motivators and barriers to the acceptability of a human milk bank among Malaysians” submitted to the Journal for review.

This is an interesting, well-written manuscript and addressed an important issue of religiously sensitive topics, especially in Islamic Countries. I see that is a good study, with a clear methodology and analysis. Discussions and conclusions are well supported by data. The limitations of the study were presented well. The abstract accurately conveys the research findings. The authors make a significant effort to make the manuscript clear.

- The last table (Table 7), consider adding a footnote containing the model statistical summary.

Thanks

Reviewer #3: Dear authors,

Congratulations for the topic of your interest and the contributes this type of research may bring for future similar public health problems. Manuscript is well writen.

6. PLOS authors have the option to publish the peer review history of their article (what does this mean?). If published, this will include your full peer review and any attached files.

Reviewer #1: No

Reviewer #2: **Yes: **Hassan Kasim Haridi

Reviewer #3: No

---

## [Editor Report · Acceptance letter]

23 Feb 2024

PONE-D-23-14900 

PLOS ONE

Dear Dr. Nik Farid, 

I'm pleased to inform you that your manuscript has been deemed suitable for publication in PLOS ONE. Congratulations! Your manuscript is now being handed over to our production team.

Kind regards, 

on behalf of

Dr. Nneoma Confidence JeanStephanie Anyanwu 

Academic Editor

PLOS ONE